# TRα2—An Untuned Second Fiddle or Fine-Tuning Thyroid Hormone Action?

**DOI:** 10.3390/ijms23136998

**Published:** 2022-06-23

**Authors:** Georg Sebastian Hönes, Nina Härting, Jens Mittag, Frank J. Kaiser

**Affiliations:** 1Department of Endocrinology, Diabetes and Metabolism, University Hospital Essen, University of Duisburg-Essen, Hufelandstr. 55, 45147 Essen, Germany; 2Institute of Human Genetics, University Hospital Essen, University of Duisburg-Essen, Hufelandstr. 55, 45147 Essen, Germany; nina.haerting@uk-essen.de (N.H.); frank.kaiser@uk-essen.de (F.J.K.); 3Institute for Endocrinology and Diabetes-Molecular Endocrinology, Center of Brain Behavior and Metabolism CBBM, University of Lübeck, Ratzeburger Allee 160, 23562 Lübeck, Germany; jens.mittag@uni-luebeck.de

**Keywords:** thyroid hormone receptor, TRα2, alternative splicing, thyroid hormone signalling, dominant-negative effect, biological role, physiological function

## Abstract

Thyroid hormones (THs) control a wide range of physiological functions essential for metabolism, growth, and differentiation. On a molecular level, TH action is exerted by nuclear receptors (TRs), which function as ligand-dependent transcription factors. Among several TR isoforms, the function of TRα2 remains poorly understood as it is a splice variant of TRα with an altered C-terminus that is unable to bind T3. This review highlights the molecular characteristics of TRα2, proposed mechanisms that regulate alternative splicing and indications pointing towards an antagonistic function of this TR isoform in vitro and in vivo. Moreover, remaining knowledge gaps and major challenges that complicate TRα2 characterization, as well as future strategies to fully uncover its physiological relevance, are discussed.

## 1. Introduction

Thyroid hormones (THs; thyroxine, T4; and the biologically active TH 3,3′,5-triiodothyronine, T3) are key regulators of organ development, growth, and cardiometabolic functions [1,2,3,4]. Circulating TH concentration is maintained by the hypothalamic-pituitary–thyroid axis through a very stable negative feedback loop [5]. The plethora of local TH effects at the cellular level is regulated by several mechanisms, including TH transmembrane transport, conversion by deiodinases, and, ultimately, binding to TH receptors (TRs) [6,7]. Canonically, TRs function as ligand-dependent transcription factors that regulate a wide set of genes in several organs [8,9,10,11]. Additionally, TRs exert physiological effects via the activation of cytoplasmic signalling pathways; this is referred to as non-canonical TR signalling [12,13,14,15,16,17]. In mammals, four major TR isoforms (TRα1, TRα2, TRβ1, and TRβ2) are encoded by two distinct genes, *THRA* and *THRB* [18]. Tissues vary in the expression levels of TRα and TRβ isoforms during development and in adulthood [19,20]. Remarkably, only TRβ1, TRβ2, and TRα1 bind T3, while TRα2 does not interact with T3 [3,21,22,23]. Nevertheless, TRα2 is evolutionarily conserved in eutherian mammals [24], and a recent study revealed high and even predominant expression of TRα2 in certain tissues [20]. Even though these observations strongly suggest a physiological function of TRα2, this remains to be proven [3,25,26]. In this review, we will focus on the molecular differences between TRα1 and TRα2, highlight the mechanisms controlling TRα2 expression, and discuss these characteristics related to a potential biological role of TRα2.

## 2. The Diverse Landscape of TRα Isoforms

### 2.1. Structural Differences of TRα Isoforms

The genetic structure of *THRA* (Chr17q21.1), the gene encoding for TRα isoforms in humans, is composed of 10 exons and has a total size of approximately 31 kb. TRα belongs to the superfamily of nuclear receptors, which share a similar domain structure and function as ligand-dependent transcription factors [27]. The activator function domain 1 (AF1) acts as a coactivator binding site and is encoded by exons 2 and 3 (Figure 1). The DNA-binding domain (DBD) is highly conserved between TR isoforms and is composed of two zinc-finger motifs that are crucial for the sequence-specific recognition of thyroid hormone response elements (TREs) on the DNA (encoded by exon 4 and 5; Figure 1). Exons 6 and 7 encode for the hinge region that contains the nuclear localization sequences (NLS) mediating nuclear import. The C-terminal ligand-binding domain (LBD) is not only necessary for T3 binding but is also involved in receptor dimerization, mediated by the ninth heptad (Figure 1), the most conserved heptad repeat in TRs, containing hydrophobic amino acids at positions 1, 5, and 8 (L367, V371, and L374) [3,28]. The ΦΦxEΦΦ sequence (LFLEVF; amino acids 400–405; Φ = hydrophobic amino acid, x = any amino acid, and E = glutamic acid) in this AF2 domain is important for coactivator and corepressor binding [3]. In the context of ligand binding, helix 12, the most C-terminal region of the receptor that carries the ΦΦxEΦΦ sequence, executes a conformational change from an ‘open’ to a ‘closed’ state that traps T3 in a hydrophobic binding pocket. The ‘closed’ conformation of helix 12 is stabilized by T3, which promotes coactivator binding, a condition generally considered as the switch mechanism. Of note, a recent study revealed that this switch is rather a coregulator shift between coactivators and corepressors bound to the TR than a ‘all or none’ switch [29]. However, a functional helix 12 is a prerequisite for TRs to act as ligand-dependent transcription factors [3,30,31].

### 2.2. The Multifold TRα Isoforms Encoded by THRA

Several TRα isoforms have been reported (TRα1, TRα2, TRα3, TRαΔ1, TRαΔ2, TRαp43, and TRαp30), which are supposedly either generated by alternative translation, alternative transcription, or alternative splicing (Figure 1) [14,32,33,34,35,36]. The expression of the truncated TRα isoforms TRαΔ1 and TRαΔ2 is presumably under the control of an internal promoter in intron 7. TRαΔ1 and TRαΔ2 had an inhibitory effect on TRα1 transcriptional activity in in vitro assays [36]. In the absence of TRα1 and TRα2 (in TRα^−/−^ mice), TRαΔ1 and TRαΔ2 are thought to severely alter intestinal development, a phenotype that is milder in TRα^0/0^ mice that additionally lack the truncated isoform [37,38,39]. Whether there is a physiological function of these truncated isoforms beyond the intestine remains highly questionable. 

The use of internal TLS (translational start sites) leads to the formation of the two larger isoforms TRαp43 and TRαp30 [35]. The translation of TRαp43 was shown to be initiated by methionine at position 39 (Met39). This isoform is most likely located in mitochondria. Thus, TRαp43 is thought to function as a mitochondrial transcription factor [40]. Mice that specifically lack the TRαp43 isoform display impaired insulin secretion and affected glucose homeostasis [41], decreased respiratory chain activity in skeletal muscle that alters muscle development and activity [42], and glucose intolerance and insulin resistance during aging [43]. In 2014, Kalyanaraman et al. demonstrated that Met150 serves as an internal TLS to produce a 30 kDa TRαp30 isoform, which is neither located in the nucleus nor in mitochondria but binds to the inner site of the plasma membrane. TRαp30 was shown to act via noncanonical signalling, activating cytoplasmic signalling pathways in primary human and murine osteoblasts, affecting the proliferation and survival of these cells [14]. However, evidence for a physiological function in vivo has not been tested to date.

Alternative splicing of the *THRA* transcript generates the two non-TH binding isoforms TRα2 and TRα3 [33,34]. TRα2 and the smaller TRα3 have a high homology with TRα1, sharing exon 1–8 and the first half of exon 9 [32,33]. While alternative splicing of TRα2 introduces the entire exon 10, an alternative splice acceptor 117 bp downstream of the splice acceptor of exon 10 is used for TRα3, resulting in a loss of the first 39 amino acids encoded by exon 10 [34]. In contrast to TRβ3, an isoform that is generated via alternative transcription and that is only present in rat [44,45], the alternative splice acceptor in exon 10 that leads to TRα3 is also present in human *THRA*. However, since the first description of TRα3, no additional studies on this isoform have been published, questioning its general biological relevance. Albeit the manifold TRα isoforms, here we will focus on TRα2 that is generated via an alternative splice site in exon 9 because the existence of this isoform was confirmed by several independent studies.

### 2.3. Alternative Splicing Controls the Intracellular TRα1:TRα2 Ratio

The transcription of TRα1 and TRα2 is under the control of the same promoter, and both TR isoforms share the first 8 exons. Thus, TRα2 shares exons 1–8 with TRα1, enabling TRα2 to interact with cofactors and bind to TREs as the AF1 and the DBD are preserved. However, the abundance of TRα2 only relies on splicing using an alternative splice site in exon 9 (Figure 1). This site is located in the middle of the ninth heptad (LLMK/VTDL) between lysine 370 and valine 371 encoded by AGG/GTG, upstream of the region encoding for helix 12 and the ΦΦxEΦΦ sequence [22]. In human *THRA*, splicing results in an exchange of 40 C-terminal amino acids of TRα1 by 120 amino acids encoded by exon 10. This leads to an increased size of TRα2 compared to TRα1, and it abrogates TH binding due to the lack of the AF2 domain including helix 12 and the ΦΦxEΦΦ sequence that are essential for a functional LBD, having dramatic effects on the function of this isoform that are explained in Section 3.

## 3. Molecular Characteristics and Function of TRα2

Although the non-T3-binding splice variant TRα2 was discovered more than 30 years ago [21,22,46], its function is still poorly understood. To exert their function as ligand-dependent transcription factors, TRs require nuclear translocation, DNA-binding ability, ligand binding, dimerization, and corepressor/coactivator binding [3,4]. The exchange of the 40 C-terminal amino acids of TRα1 by 120 amino acids specific to TRα2 not only impairs T3 binding [21,22,23] but also results in the disruption of the ninth heptad within the LBD required for heterodimerization with retinoid X receptor (RXR) [28,47], thus potentially altering the recruitment of the receptor to the DNA [48].

### 3.1. DNA Binding and Dimerization Characteristics of TRα2

The DNA target sites bound by TRs are referred to as TREs, of which different types have been described. TREs consist of two half-sites containing the consensus-motif AGGTCA or variations of it [18,49]. Depending on the orientation of the two half-sites and the number of nucleotides spacing them, TREs are classified as direct repeats (e.g., DR4: AGGTCAnnnnAGGTCA) [50] or palindromic arrangements (e.g., PAL0: AGGTCA TGACCT) [51] that can be inverted or everted (e.g., IP6: AGGTCAnnnnnnTGACCT; ER6: TGACCCnnnnnnAGGTCA) [18,52]. TRα2 binds to TREs with a substantially lower affinity than TRα1 [21,53,54], although they share identical DBDs. Moreover, TRα2/TRE complexes are less stable compared to TRα1/TRE [55]. The binding affinity of TRα2 varies among different TRE types: the binding of TRα2 to palindromic and inverted palindromic TREs is impaired [48,56], while binding to direct repeat TREs with 4 bases of spacing (DR4) is detectable but restricted to a subset of octameric DR4s [48,57], with a preference for certain spacer motifs and the presence of perfect AGGTCA consensus TRE half-sites [55]. Of note, the described TRE binding characteristics of TRα2 are based on testing selected TREs of each subtype, but an analysis at a whole-genome level is required to test whether these characteristics hold true in vivo. The low DNA binding affinity of TRα2 has been attributed to its specific C-terminus as the deletion of this region enhances the binding of TRα2 to palindromic TREs [54]. The altered C-terminus of TRα2 could directly interfere with DNA binding but also indirectly compromise DNA binding by affecting the dimerization properties of TRα2 [58].

While TRα2 forms homodimers, the ability to interact with other TR isoforms in heterodimers is questioned in some studies due to disruption of the ninth heptad region [54,59,60]. Consistent with this notion, TRα2 was found to be unable to interact with RXR in vitro in the absence of DNA unless the ninth heptad was restored [48]. However, other studies demonstrated the capability of TRα2 to form heterodimers with RXR when bound to DR4 TREs [48,55,56,57], suggesting that DR4 binding mediates the interaction of TRα2 with RXR. In contrast, in the case of palindromic TREs, the reconstitution of the ninth heptad is required for binding by TRα2 as a heterodimer with RXR [61], indicating that TRα2 cannot bind to these TREs unless recruited by RXR [48]. However, the requirement of reconstitution of the ninth heptad appears to be sequence-independent to allow binding of TRα2 to inverted palindromic TREs, although it does not allow DNA-independent heterodimerization with RXR, suggesting that for these TREs, the ninth heptad rather serves as a spacer that separates the C-terminus of TRα2, which otherwise inhibits DNA binding [48] and therefore dimerization.

### 3.2. Posttranslational Regulation of TRα2

Interestingly, posttranslational modifications also affect the DNA binding of TRα2. The dephosphorylation of two serine residues (S474-S475) in the C-terminal region of rat TRα2 has been found to enhance DNA binding, as shown by the enhanced TRE binding of a TRα2-SA mutant that lacks the phosphorylation sites [62]. In agreement with this finding, the monomeric and homodimeric binding of rat TRα2 to DR4 and inverted palindromic TREs in vitro was enhanced if TRα2 was expressed in bacteria and thus was non-phosphorylated. The in vitro translation of TRα2 with rabbit reticulocyte lysates resulted in phosphorylated TRα2 and subsequently decreased TRE binding [48,62]. Of note, the specific serine residues (S474-S475) are absent in the human TRα2 isoform. Nevertheless, it is suggested that other serine residues nearby might be phosphorylated instead because the TRE binding affinity of the human TRα2 isoform has been reported to be affected by phosphorylation as well [62]. However, no data were shown to confirm this finding. Interestingly, apart from affecting DNA binding, phosphorylation controls the subcellular localization of TRα2 as well. While unphosphorylated TRα2 is mainly found in the nucleus, the phosphorylated isoform accumulates in the cytoplasm [63]. Remarkably, this seems to be a TRα2-specific function. Due to the different C-terminus TRα2 also lacks the NES in helix 12, one of two NES that are present in TRα1, while both NLS are preserved in TRα2. Thus, the loss of one NES rather suggests a mainly nuclear localization of TRα2. However, the phosphorylation of TRα2 at its C-terminus can compensate for that loss. Whether phosphorylation might serve as a molecular switch to regulate its function yet needs to be determined. 

Moreover, sumoylation of TRs at lysine (K) residues affects the interaction of TRs with other transcription factors, as well as gene induction and repression [64]. TRα1 has two main sumoylation sites at K283 and K389 and an alternative sumoylation site at K288 that can compensate for K283 [65]. While K389 lies within the TRα1-specific region of the C-terminus, K283 and K288 are located within a region shared by both isoforms. Thus, K283/K288 might be sumoylated in TRα2 as well. However, further analyses are required to dissect isoform-specific sumoylation sites and establish whether they affect TR functions. 

### 3.3. Corepressor Binding of TRα2

Apart from TRE binding and dimerization, interactions with corepressors constitute most likely the biggest difference between TRα1 and TRα2. Corepressors such as the nuclear receptor corepressor (NCoR) and the silencing mediator of retinoic acid receptor and TRs (SMRT) are essential to repress the transcription of T3-positively regulated genes and hence the physiological function of TRs [66,67,68,69]. For TRα2, only a weak or even no interaction with NCoR or SMRT was found [55,70,71]. As shown for limited TRE binding and heterodimerization, the lack of a complete ninth heptad region also plays a role in the interaction of TRα2 with corepressors because restoring the ninth heptad greatly enhances the interaction with corepressors [70,71]. Ligand-independent gene repression is markedly impaired in mice expressing a mutant TRβ (R429Q) that cannot effectively recruit NCoR [72]. Thus, the ligand-independent repression of TRα2 must be attenuated compared to a TR isoform that can interact with NCoR to repress the expression of target genes. 

### 3.4. Molecular Functions of TRα2 in TH Signalling

Several publications demonstrated a weak dominant-negative effect (DNE) of TRα2 on TRα1- and TRβ-mediated target gene transactivation in vitro [23,33,48,55,56], potentially to fine-tune TH signalling. Not only TRs but also other hormone receptors such as RXR and the estrogen receptor were shown to be inhibited by TRα2 [58]. Despite the demonstration of this DNE, the underlying mechanisms are not well understood.

On a molecular level, the antagonistic function of TRα2 is related to its C-terminus. Experiments with hybrid receptors containing the TRα2 C-terminus revealed that the DNE of TRα2 is transferable to other receptors [58]. Mechanistically, the DNE of TRα2 may be exerted by competitive binding of TREs, or by a DNA-binding independent mechanism such as the competitive binding of cofactors or the formation of inactive heterodimers [73,74] (Figure 2). 

Several findings support the hypothesis that the dominant-negative activity of TRα2 is exerted by competitive TRE binding, although the DNA binding affinity of TRα2 is lower than that of TRα1. First, mutations enhancing DNA binding also enhance the DNE of TRα2 [59]. Moreover, the low DNA binding affinity of TRα2 is enhanced by dephosphorylation, which could display a posttranslational control mechanism to adjust the DNE of TRα2 [62]. In addition, a higher expression of TRα2 than TRα1, which has been found in several murine tissues [20], could also compensate for the low DNA binding affinity. Furthermore, TRα2 inhibits the action of chimeric transcription factors only when they contain the N-terminus of TRα1 and bind to TREs, but not if they contain the C-terminus of TRα1 and bind to other response elements, supporting the hypothesis that inhibition by TRα2 is DNA-dependent [59]. Investigations to clarify the requirement of DNA binding of TRα2 for its antagonistic function using TRα2 variants with a mutated or deleted DBD revealed contrasting findings [48,73], and some studies suggest that the requirement of DNA binding is TRE-dependent [48,56]. 

These findings suggest that DNA-independent mechanisms play a role for the DNE as well. TRα2 was demonstrated to inhibit only the transactivation of positive TREs, which are located within regulatory regions of genes upregulated in response to T3 [26], supporting the hypothesis that it competitively binds a factor required for TR-mediated transactivation [74]. More recent data suggest that TRα2 can inhibit both positively and negatively regulated TREs in vitro; noteworthily, the overexpression of TRα2 in vivo had no significant effect on negative TREs [75]. However, this may also explain why its antagonistic effect is limited and further strengthens the hypothesis that the repression of T3-mediated gene expression is attenuated due to a lack of corepressor binding, as shown for the TRβ mutant [72].

The reconstitution of the ninth heptad in TRα2 greatly enhances the DNE, as on the one hand it allows heterodimerization with other TRs and on the other hand augments DNA binding [48]. These Janus-faced results complicate drawing conclusions regarding the role of DNA binding or the sequestration of cofactors for the DNE of TRα2 [48]. Moreover, the C-terminal domain of α2 was demonstrated to be sufficient for the inhibition of transactivation mediated by the estrogen receptor without requiring the N-terminal DBD [58], but it does not inhibit the binding of TRα1 to TREs [59]. Additionally, a recent study revealed a mutation in a patient, affecting both TRα1 and TRα2 [76]. This point mutation is located outside the DBD and enhances the DNE of TRα2, suggesting enhanced cofactor binding rather than affecting DNA binding [76].

These diverse findings regarding the relevance of DNA binding raise the question of whether there might be different TRE-dependent mechanisms involved in exerting the DNE of TRα2 [48]. In conclusion, the DNE of TRα2 could be modulated separately for different TRE types and diversely by different mechanisms, which would further specify the fine-tuning of TH signalling. 

## 4. Mechanisms Controlling Expression of TRα2

### 4.1. Tissue-Specific Expression of TRα2

Even before the discovery of TRs, differences in nuclear bound T3 in several tissue samples suggested the tissue-specific content of ‘T3 binding sites’ [77], today better known as TRs. The first studies used northern blot analysis and in situ hybridization to investigate the tissue-specific expression of TR transcripts [78,79,80,81]. The low amount of TR transcripts, however, makes detection by in situ hybridization quite challenging. Nevertheless, these techniques enabled the identification of tissues predominantly expressing TRα or TRβ, as well as crude TR isoform-specific differentiation. TRα1 is predominantly expressed in heart, brain, adipose tissue, skeletal muscle, and bone, whereas TRβ1 is the predominant isoform in liver and TRβ2 in pituitary. TRα2 transcripts were identified most abundantly in brain and heart [81]. Of note, as a limitation, the direct comparison of these studies may not be possible as different species such as chicken [79], rat [78,80], or pig were investigated, and, moreover, not all studies differentiated between the different TR isoforms [79,82]. Due to a lack of suitable antibodies specifically detecting different TR isoforms, all studies investigating TR protein amount must be treated with caution. For instance, TRα2 protein was detected in human, dog, and guinea pig hearts but it was absent in rat and mouse hearts [83]. Contrary to this, a recent study using mouse models with endogenously tagged-TRs revealed TRα2 in brain and heart [20]. Remarkably, the expression of TRα2 was higher than that of TRα1. Extensive studies of TR-knockout and TR-mutant mouse models gave genetic proof of the tissue-specific expression of the different TR isoforms [38,84,85,86,87,88,89,90]. However, the underlying mechanisms that orchestrate the tissue-specific expression of TRα isoforms are yet to be determined. 

### 4.2. Factors That Influence the TRα1:TRα2 Ratio

Alternative splicing of *THRA* controls the intracellular ratio of TRα1:TRα2 and can ultimately and very rapidly adjust cellular TH-responsiveness [33,77,91]. Interestingly, several factors are known to affect the TRα1:TRα2 ratio. Fasting resulted in a three-fold increase of TRα2 over TRα1 in rat livers, while the TRα1:TRα2 ratio in heart was unaffected [92]. Further, pharmacological T3 treatment altered the balance of TRα1 and TRα2 in favour of TRα2 in HepG2 cells, possibly by changing the ratio of two splicing factors (hnRNP A1 and SF1) that are involved in alternative splicing of TRα2 [91,93]. An isoform switch to non-T3 binding TRα2 suggests the protective adaption of cells against excessive TH-mediated gene expression [93]. Additionally, low T3 serum concentrations correlated with a higher TRα1:TRα2 ratio in livers of critically ill patients. Of note, here splicing factors seemed to be not involved [94]. Fasting and T3 both induce the expression of the PPARγ coactivator α (PGC-1α) in liver [95,96], a transcription factor that also acts on mRNA processing [97,98]. In HepG2 cells, the overexpression of PGC-1α resulted in a decreased TRα1:TRα2 ratio, thus providing a unifying explanation of increased TRα2 expression in fasting and under high T3 concentration [99]. However, the underlying mechanisms that regulate the alternative splicing of TRα2 are manifold, complex, and not yet fully understood. 

### 4.3. Cis-Regulatory Factors of TRα2 Alternative Splicing

Several sequences in intron 9 (splicing enhancer α2, SEα2; TR-intronic splicing enhancer 3, TR-ISE3) and exon 10 (G-rich element, G30; exonic splicing enhancer in exon 10, ESX10) of *THRA* that are necessary for the accurate splicing of TRα2 have been identified [24,100,101,102]. The SEα2 sequence is located directly downstream of the stop codon of TRα1 and is about 80 bp long. Remarkably, even though this sequence is located in the 3′-UTR of TRα1, it is highly conserved among mammals (>95% homology), and genetic variants in this sequence dramatically reduce the alternative splicing of TRα2 [91,100]. Two mouse models further support the importance of SEα2 for the splicing of TRα2. In the first model, TRα1 was tagged with GFP at the C-terminus to overcome a lack of suitable TRα antibodies. This model enabled one to study temporospatial TRα1 expression during brain development and revealed an exclusively nuclear localization of TRα1. The second model is the TRα2 knockout mouse, in which a SV40-polyA/neomycin cassette was introduced directly downstream of exon 9 [103]. However, in both models, the fusion of GFP to TRα1 and the introduction of the SV40-polyA/neomycin cassette extended the distance between the alternative splice site and SEα2, consequently fully abolishing the expression of TRα2. Of note, the loss of alternative splicing of one TRα2 allele resulted in an increase of TRα1 level, showing that the TRα1:TRα2 ratio is under the control of this alternative splicing mechanism [103,104]. 

Three additional sequences that are important for the alternative splicing of TRα2 are TR-ISE3, ESX10, and G30. TR-ISE3 is located upstream of the 3′-splice site of TRα2 and affects splicing additively and independently of SEα2 [101]. ESX10 consists of several 42–43 bp long sequences promoting the splicing of TRα2 [101]. In contrast, the G30 sequence inhibits the splicing of TRα2. This inhibitory effect correlates with the G-content of the sequence as mutations that increase the number of G-clusters strongly decrease splicing. Thus, the involvement of a G-quadruplex, a noncanonical secondary nucleic acid structure that was shown to affect transcription, translation, polyadenylation, and splicing, was suggested as a possible *cis*-regulatory element in the alternative splicing of TRα2 [102]. 

### 4.4. The Antisense Overlap with Rev-erbα—Regulatory Function and Genetic Evolution

With respect to TR-ISE3, G30, and especially for ESX10, the antisense overlap with *NR1D1* encoding for Rev-erbα must be mentioned. This special genomic organization results in a bidirectional coding sequence of 200 bp for exon 10 of TRα2 and exon 8 of Rev-erbα, including ESX10 (Figure 3A) [101,105,106]. G30 is directly antisense downstream of the stop codon of Rev-erbα, thus mapping to the 3′-UTR of Rev-erbα as well as TR-ISE3 [101,102]. The higher expression of Rev-erbα is correlated with an increase of the TRα1:TRα2 ratio. Possible mechanisms such as interference with exon 10 transcription, the destabilization of TRα2 mRNA, or the inhibition of TRα2 splicing have been suggested. Ultimately, the inhibition of TRα2 splicing by Rev-erbα mRNA, which is complementary to the mRNA sequence of exon 10 of TRα2, is the most likely explanation [106,107,108]. Base pairing between Rev-erbα and TRα2 mRNA might prevent the binding of splicing factors [109], and TR-ISE3 is suggested to function as a possible initiator of this base pairing [101].

Even though the exact molecular mechanisms remain unknown, the regulation of TRα2 splicing by the antisense overlap of Rev-erbα and TRα2 is a physiological function that can only be found in mammals [24,102]. Here, the proximity between *THRA* and *NR1D1* is the closest, followed by birds where the sequences are only 2.8–3.8 kb apart, whereas in amphibians and reptiles the distance is about 9–15 kb [102] (Figure 3B). No sequence corresponding to TRα2 is present in *Xenopus tropicalis* [101]. In conclusion, phylogenetic analysis suggests that TRα2 originated with the infrequent alternative splicing of a read-through transcript, enabled by the antisense overlap of TRα2 and Rev-erbα coding sequences, in a common ancestor of marsupials and eutherian mammals [102]. Marsupials are the only mammals that have lost TRα2, probably due to an absence of positive selection [102], but the evolution and conservation of TRα2 in eutherian mammals strongly suggests a physiological function of this isoform. 

## 5. Unravelling the Biological Function of TRα2—Lessons of Mice and Men

### 5.1. Deducing TRα2 Function from Expression and In Vitro Experiments

The physiological relevance of TRα2 is still questioned [26], even though evolutionary conservation and the presence of the protein in most tissues point towards a specific function of the isoform [20,102]. As rapid changes in TH-dependent TRα1 action are crucial for murine development [110], the switch from TRα1 towards TRα2 splicing may represent a mechanism to regulate quickly transcript levels of TRα1 without requiring a specific function of the TRα2 protein. However, the half-life of TRα2 mRNA, which has been found to be longer than that of TRα1 mRNA [106], and the presence of the protein in most tissues, in some cell types even with a higher abundance than TRα1, indicate a specific function of TRα2 [20]. One potential function of TRα2 shown in vitro is antagonizing TRα1 and TRβ action as a possible fine-tuning mechanism of TH signalling [23,33]. In general, as the DNE of TRα2 appears to be weak, excess TRα2 protein is required to achieve a substantial inhibition of TRα1 function in vitro [55,59,74], potentially related to the low DNA binding affinity [59], the incapability of TRα2 to interact with CoRs [55], and the lack of a complete ninth heptad [48]. However, the comparably weak inhibition of TH signalling may be important to gradually finetune TR action rather than to impair it drastically, supporting the hypothesis that TRα2 plays a role in vivo. 

### 5.2. Lessons of Mice and Men

A first attempt to determine a possible physiological role of TRα2 was the generation of a TRα2 knockout mouse model [103]. Unfortunately, the abolishment of TRα2 led to the increased expression of TRα1 with a mixed hypothyroid/hyperthyroid phenotype, thus making it difficult to attribute the phenotype solely to the loss of TRα2. However, the results suggest that the TRα1:TRα2 ratio provides a fine-tuning mechanism controlling growth and homeostasis in mammals [103]. Furthermore, a knockout of both TRα isoforms in mice exhibits higher sensitivity towards T3 in tissues that are predominantly controlled by TRβ, likely due to the absence of TRα2-mediated inhibition [111]. The homozygous TRα1-GFP mice, in which TRα2 expression was abrogated, only had a mild phenotype compared to wild-type mice, and the determined differences were rather related to a mildly impaired ability of TRα1-GFP to repress TH-target gene expression than to the loss of TRα2 [104]. Indirect indications of the physiological relevance of TRα2 were concluded from studies using *Pax8^−^*^/*−*^ mice. Homozygous *Pax8^−^*^/*−*^ mice die during the first weeks of life due to impaired TH production leading to a strong repressive effect by the apo-TRs [112]. Of note, these mice can be rescued by the inactivation of TRα but only if both isoforms are knocked out, thus concluding an antagonizing effect of TRα2 on the remaining TRβ in the *Pax8^−^*^/*−*^/TRα1^−/−^ mice that is lethal in absence of TH [113].

In the human heart, a shift from TRα1 to TRα2 expression was observed in heart failure [114,115], and elevated TRα2 expression was found to attenuate TRα1-mediated hypertrophy in this condition [116]. These findings suggest an antagonistic activity of TRα2 that may play an important role in the pathophysiology of heart disease. Furthermore, recent findings reveal a potential relation of TRα2 and cancer. Low TRα2 levels are associated with unfavourable tumour characteristics and an increased mortality or lower disease-free survival in breast cancer, as demonstrated in several studies [117,118,119]. However, this association was not found to be independent of other prognostic factors in every case [118,119]. Moreover, other studies reveal partially contradictory results [120]. Thus, further investigation is required to disclose the role of TRα2 in cancer.

More indications for the potential physiological relevance of the DNE of TRα2 are displayed in patients with genetic variants affecting both TRα isoforms: a patient with a mutation between the DBD and LBD of both TRα1 and TRα2 shows a partially hypothyroid phenotype, potentially caused by the enhanced DNE of TRα2 [76]. Consistent with this finding, it has been suggested that the TRα2^N359Y^ variant affecting both isoforms identified in a patient with an atypical phenotype of resistance to thyroid hormone could increase the T3 sensitivity in TRβ1-dominated tissues, contributing to the observed phenotype [121]. To date, no pathogenic *THRA2*-specific variants associated with a distinct phenotype that could further contribute to the understanding of the biological function of TRα2 have been described. 

## 6. Future Challenges to Define the Biological Role of TRα2

The poor characterization of TRα2 function is partly due to the lack of isoform-specific antibodies. Moreover, attempts to study the consequences of TRα2 knockout in mice have been complicated by the concomitant overexpression of TRα1 [103]. A TRα2 knockout mouse model that maintains a normal TRα1 expression level could provide more insights into the physiological role of TRα2 but could only give indirect evidence for its function, and the presence and effect of truncated protein variants due to the knockout needs be carefully evaluated [25]. 

A possible solution to allow for the specific analysis of the different isoforms would be the use of tagged isoforms enabling the specific detection of isoforms with tag-specific antibodies. This strategy has already been followed by Shabtai et al., where TRβ has been tagged to study TH-dependent coregulator interactions [29]. Moreover, interesting findings regarding the expression pattern and ratio of the two TRα isoforms were demonstrated by Minakhina et al., where TRα has been tagged N-terminally with a 2xHA tag [20]. Further research needs to follow to uncover isoform-specific DNA binding profiles and protein interactions, as well as the mechanism regulating isoform splicing to understand the role of TRα2 during development and for different tissues. Moreover, intriguing findings showing phosphorylation-dependent DNA binding affinity and subcellular localization [62,63] representing a potential mechanism to regulate TRα2 function need to be confirmed and further analysed on the endogenous level. The finding that TRα2 can also be located in the cytoplasm [63], and the demonstration of the noncanonical functions of TRs [15], raises the question of whether TRα2 might have non-genomic functions as well.

Taking advantage of tagged TR isoforms may represent a promising approach to further investigate the role of the enigmatic TRα2 isoform endogenously in vivo. Moreover, to fully unravel the function of TRα2, new models are needed. Considering that the nuclear-cytoplasmic shuttling of TRα2 can be modulated by the phosphorylation of specific residues of the C-terminus, introducing a phosphomimetic mutation could ban TRα2 from the nucleus, thus abrogating its DNE on TRα1-mediated gene expression. Further, such a model could also shed light on the potential noncanonical actions of TRα2. Likewise, animal models with specific point mutations in TRα2 could be helpful, even if only to exclude a contribution of TRα2 to phenotypes associated with TRα1 defects. Of note, the use of iPSCs (induced pluripotent stem cells) instead of mouse models could not only accelerate such studies but also allow for the investigation of *THRA* mutations in the genetic background of patients, facilitating genetic manipulation and retaining the possibility to study the role of TRα2 in various cellular models. However, the further identification and basic characterization of functional motifs that discriminate TRα2 from TRα1 precede any generation of an iPSC or even a mouse model. 

## Figures and Tables

**Figure 1 ijms-23-06998-f001:**
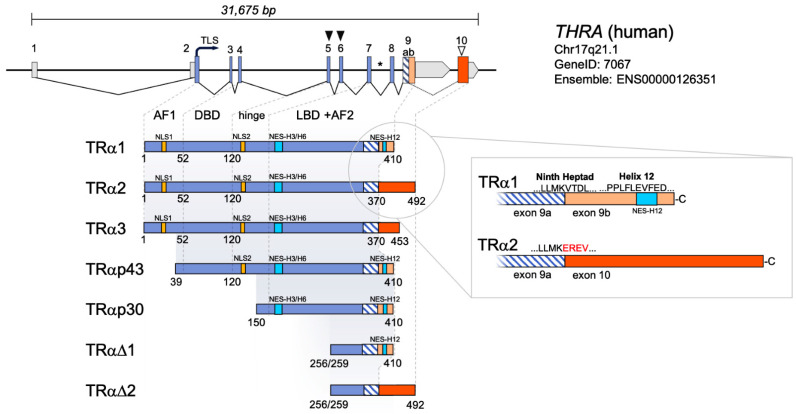
Genetic organization of *THRA* and domain structure of TRα isoforms. *THRA* consists of 10 exons encoding for a diverse set of TRα isoforms. Translation of TRα1, TRα2, and TRα3 starts at a translational start site (TLS) in exon 2, whereas the use of two alternative TLS (closed triangles) results in generation of TRαp43 and TRαp30. TRαΔ1 and TRαΔ2 originate from alternative transcription at an internal transcriptional start site (asterisk) in intron 7. Generation of TRα2 and TRα3 also requires alternative splicing at an alternative splice site located in exon 9. The alternative splice acceptor for TRα3 is located in exon 10 (open triangle). This results in several truncated isoforms lacking one to several domain structures such as the activator function 1 domain (AF1), the DNA-binding domain (DBD), the hinge region, parts of the ligand-binding domain (LBD), and the C-terminal AF2 domain. Intracellular localization is also affected by the different domain structures as loss of a certain domain also leads to loss of nuclear localization sequences (NLS; yellow boxes) and nuclear export sequences (NES; light blue boxes). The box displays a zoom in exon 9 and 10. An alternative splice site in exon 9 divides this exon into two parts (exon 9a and b), whereas splicing of exon 9a with exon 10 results in the formation of TRα2, consequently disrupting the ninth heptad sequence as well as helix 12 and the NES-H12 in this region.

**Figure 2 ijms-23-06998-f002:**
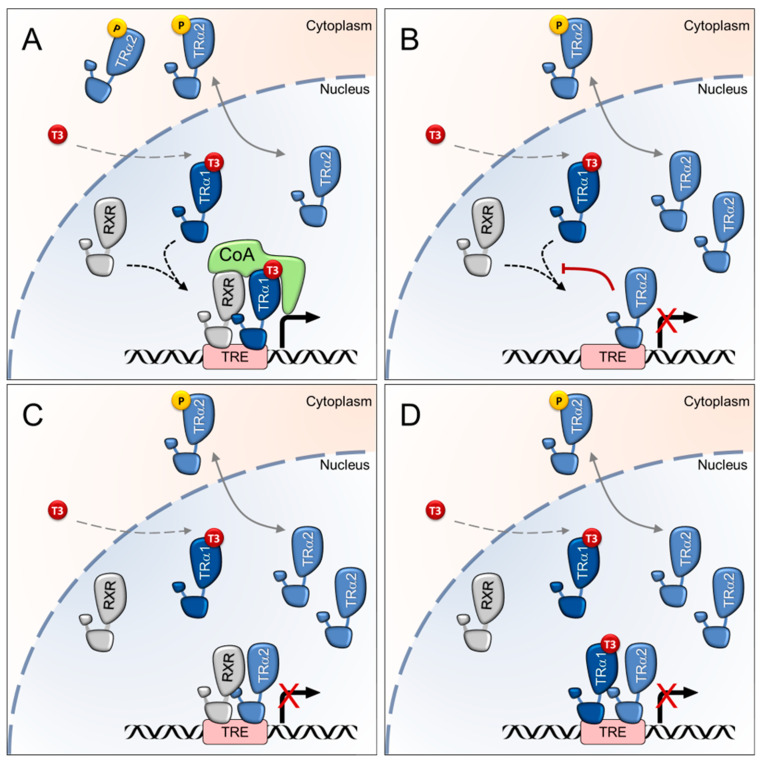
Proposed mechanisms underlying the dominant-negative activity of TRα2. (**A**) Canonical action of TRα1 in the absence of excess nuclear TRα2. After entering the nucleus, the biologically active TH 3,3′,5-triiodothyronine (T3) interacts with TRα1, which binds to TREs (TH response elements) of TH-target genes, preferentially as a dimer with other hormone receptors as RXR (retinoid X receptor). If low levels of TRα2 are present in the nucleus, TRα1 exerts its genomic function by recruiting coactivators (CoA) to stimulate the expression of target genes in the presence of T3. (**B**) While phosphorylated TRα2 (indicated with P) is located in the cytoplasm, unphosphorylated TRα2 accumulates in the nucleus and potentially antagonizes TH signalling by competitive binding of TREs impeding binding of TRα1. (**C**) Competitive DNA-dependent sequestration of cofactors as RXR required for TRα1 action, or (**D**) formation of inactive TRα1/TRα2 heterodimers, can alternatively hamper TRα1-mediated transactivation of target gene expression.

**Figure 3 ijms-23-06998-f003:**
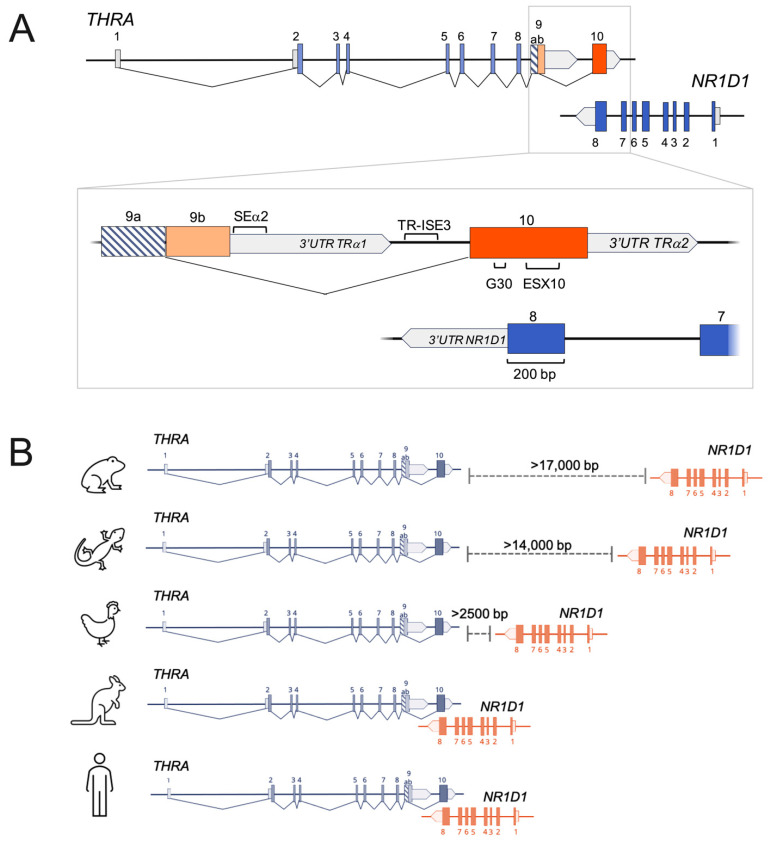
Alternative splicing of *THRA* and overlap with *NR1D1*. (**A**) Antisense overlap of *THRA* with *NR1D1* results in a 200 bp-long bidirectional coding sequence in exon 10 of *THRA* and exon 8 of *NR1D1*. Several sequences important for alternative splicing of *THRA* have been identified. The splicing enhancer α2 (SEα2) is located directly downstream of exon 9 in the 3′ untranslated region (UTR) of TRα1, while the intronic splicing enhancer TR-ISE3 lies between the 3′-UTR of TRα1 and exon 10 of TRα2. The guanine-rich element G30 and ESX10 (exonic splicing enhancer in exon 10) map directly to the coding sequence of exon 10. (**B**) Evolution from distal convergent transcription in non-mammals (e.g., amphibians and reptiles) to bidirectional overlapping transcription in mammals (e.g., marsupials and humans).

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
