# Peer review of "TRα2—An Untuned Second Fiddle or Fine-Tuning Thyroid Hormone Action?"

_ijms, 2022, doi:10.3390/ijms23136998_

Round 1

Reviewer 1 Report

Thyroid hormone receptors are encoded by two genes generating different isoforms. This review of the literature focuses more specifically on the TRα2. While there are few reviews of the literature on this isoform, the authors present the state of knowledge of the molecular characteristics, the mechanisms of regulation of the expression and the specificities of action of TRα2. However, to truly assess the biological role of TRα2, data on the role of this receptor in human physiology must be added. In particular, the review could be supplemented with more recent data on the link between TRα2 and cancers and the notion of using iPS cells to study the biological role of TRα2 must be expand.

The overall organization needs to be reviewed to clarify the message:

- present which exon code for which functional domain of the protein before describing the different specificities of the different isoforms.

- present the different functional domains and their role in the mode of action of TRα1 before describing the differences between the two isoforms

- the paragraph from line 273 to 280 should be added to the paragraph “The diverse landscape of TRα isoforms”

- Figure 3A should be mentioned instead in the manuscript because it helps to understand how from the same gene, different isoforms are produced.

In addition, there are repeated concepts and missing definitions: line 49 “alternative translational start sites (TLS)”, line 97 “ninth heptad”, “φφxEφφ”. In addition, a schema illustrating the different cis-regulatory factors would clarify the discourse.

unscientific term: “efforts to extract”

Questions :

-          What is the TRHA gene locus?

-          What are the different types of TRE? What is positive and negative TRE? The addition of a schema would be appreciated.

-          Line 45: the title “The multifold TRα isoforms encoded by THRA” is redundant with the section title “The diverse landscape of TRα isoforms”.

-          The authors indicate that the TRβ isoform is the predominant isoform in the liver (Line 258). However, the paragraph on “Factors that influence the TRα1:TRα2 ratio” mentions mainly results in the liver. Can the authors explain this?

Figure 1:

-          Is TLS symbolized by TLS or closed triangles? How many of these exist in the human THRA gene?

-          Where is the alternative splice site generating TRα2 and TRα3?

-          What is the legend of the insert?

Figure 2:

-          A - What is the role of TR α2? How does it carry out its dominant-negative activity?

-          B,C,D - More detail in the legend to understand the schemas.

Reviewer 2 Report

The review by Georg Sebastian Hönes and co-workers accurately brings together the knowledge of a little-regarded isoform of the thyroid hormone receptor alpha.

But the interest in this review also lies in the integration of this knowledge to discuss the function of this isoform.

I only have to propose minor revisions

1.     Line 51-54, although this is not the subject of the review, it seems to me that a sentence could be added to point out the role of the truncated forms suggested from the TR KO data.

2.     Line 71, a sentence could be added to report the knockout result of the p43 form.

3.     Line 74, the authors indicate that there is little evidence on the function of the p30 isoform. But what are these evidences?

4.     To date, the absence of demonstration of TRa3 in species other than the rat (line 80) could suggest that it is a form specific to this species as for the TRb3 form. It could be reported.

5.     Line 102, attention should be paid to recent results that discuss the switch mechanism (Shabtai et al, 2021 Gene&Dev 35:367).

6.     I note the absence of data on the role of the A/B domain (AF1) which in the TRa2 isoform is the only part of the receptor which has a transactivation function.
